# The Impact of Laparoscopic Surgery on Fertility Outcomes in Patients with Minimal/Mild Endometriosis

**DOI:** 10.3390/jcm13164817

**Published:** 2024-08-15

**Authors:** Georgios Grigoriadis, Horace Roman, Fani Gkrozou, Angelos Daniilidis

**Affiliations:** 11st University Department in Obstetrics and Gynecology, Papageorgiou General Hospital, School of Medicine, Aristotle University of Thessaloniki, 56429 Thessaloniki, Greece; drgeorgiosgrigoriadis@gmail.com; 2Institut Franco-Europeen Multidisciplinaire d’Endometriose (IFEMEndo), Endometriosis Centre, CliniqueTivoli-Ducos, 33000 Bordeaux, France; horace.roman@gmail.com; 3Franco-European Multidisciplinary Endometriosis Institute (IFEMEndo), Middle East Clinic, Burjeel Medical City, Abu Dhabi 7400, United Arab Emirates; 4Department of Obstetrics and Gynecology, Aarhus University, 8210 Aarhus, Denmark; 5Department of Obstetrics and Gynecology, University General Hospital of Ioannina, 45500 Ioannina, Greece; fani.gkrozou@gmail.com

**Keywords:** endometriosis, infertility, laparoscopy

## Abstract

Minimal/mild endometriosis (MME) is independently associated with reduced fecundity rates. In this review article, we discuss the role of laparoscopic surgery in enhancing the fertility outcomes of patients with MME. Laparoscopic management of MME enhances fecundity and increases the chances of spontaneous conception in appropriately selected cases. However, laparoscopy cannot be routinely recommended in asymptomatic patients with the sole purpose of diagnosing and treating potentially present MME. Equally, and based on existing information, the laparoscopic management of MME cannot be routinely recommended prior to in vitro fertilisation (IVF) attempts due to the lack of robust and beneficial evidence. Because an overlap between unexplained infertility and MME cases likely exists, the development of reliable, widely available, non-invasive tests for the diagnosis of MME may revolutionise the management of cases currently classified as unexplained infertility. In a disease as diverse as endometriosis, management decisions should be based on a multitude of factors. Future studies should focus on reporting the outcomes of interventions for MME on fertility and obstetric outcomes, clearly differentiating between disease stages and phenotypes.

## 1. Introduction

Endometriosis is a benign gynaecological disease of unknown aetiology which affects 1 in 10 women of reproductive age [1]. It is characterised by the presence of ectopic, endometrium-like tissue leading to an oestrogen-dependent, chronic inflammatory process [2], and is commonly linked with pelvic pain and/or infertility. Fecundity rates in couples of reproductive age with no documented infertility are estimated to be between 15% and 20%, whereas in those with untreated endometriosis, rates vary from 2% to 10% [3]. Endometriosis is a heterogeneous disease and three phenotypes of the disease, that may co-exist, are recognised: superficial peritoneal endometriosis (SUP), deep endometriosis (DE), and ovarian endometrioma (OMA) [1].

The commonly used 1996 revised American Society for Reproductive Medicine (rASRM) endometriosis classification system [4], recognises four stages of the disease: minimal (stage I), mild (stage II), moderate (stage III) and severe (stage IV) endometriosis. However, the correlation between an rASRM stage and the reproductive outcome following conservative endometriosis surgery has been shown to be poor [5], and other tools, such as the Endometriosis Fertility Index (EFI), have shown a more satisfactory performance in predicting the chances of natural conception [6]. Despite its apparent limitations [5,7,8], rASRM remains commonly used worldwide. In spite of advances in non-invasive testing [9], as well as novel imaging applications [10], the gold standard in the diagnosis of stage I/II endometriosis [or minimal/mild endometriosis (MME)] remains diagnostic laparoscopy, which offers the additional benefit of a histological confirmation of the diagnosis.

MME is believed to be present in 15 to 50% of patients suffering from endometriosis [11], while the appearance of lesions can vary greatly. Nisolle et al. recognise 3 types of SUP lesions [12]: red lesions, which represent the initial stages of the disease; black lesions, representing the second stage; and white, quiescent lesions. Endometriosis may also present as pelvic peritoneal defects (Allen-Masters syndrome) [13]. Histological evidence of endometriosis (occult endometriosis) may be found in the biopsies of clinically negative peritoneum from patients with pelvic pain [14], although the clinical significance of occult endometriosis in the context of pain or infertility remains unclear.

Whether there is an actual link between the presence of MME and infertility has been a matter of debate for many years [15,16,17]; recent evidence suggests that it is independently associated with infertility [18]. Women with MME may have a lower probability of pregnancy compared to those with unexplained infertility and a normal pelvis [19], while the laparoscopic management of MME in infertile women may enhance fecundity [20,21,22].

The aim of this review article is to present the available evidence on the mechanisms underlying MME-associated infertility and discuss the role of laparoscopic surgery in the management of infertility secondary to MME. For the purposes of this article, MME is considered as a single entity and the term is used interchangeably (as a proxy) with rASRM stage I/II endometriosis and SUP.

## 2. Results

### 2.1. Mechanisms of Infertility in Patients with Minimal/Mild Endometriosis

Reduced fecundity rates in women with MME may be a result of various mechanisms: The disease has been linked with an increased risk of moderate to intense deep dyspareunia [18], which may be responsible for reduced coital frequency.

The proinflammatory microenvironment that exists within the MME lesions is likely to play a key role in MME-associated infertility: Endometriotic lesions exhibit an increased production and release of chemokines, cytokines and prostaglandins [23], resulting in chronic inflammation. There is increasing evidence of an association between chronic inflammation and oxidative stress [24], which leads not only to infertility, but also to endometriosis disease progression [25]. Oxidative stress occurs as a result of an imbalance between reactive oxygen species (ROS) and the intrinsic anti-oxidant mechanisms [26]. Abnormally high levels of ROS, as a result of altered iron metabolism leading to higher levels of iron, ferritin and haemoglobin in the peritoneal fluid of women with endometriosis [25], can have a negative impact on oocyte quality and ovarian reserve [27], resulting in “oocyte aging” [26], as well as impairing implantation and early embryonic development [28]. High levels of activated macrophages, growth factors and activated cytokines in the peritoneal fluid exert a toxic effect on sperm [29].

Furthermore, Leyendecker et al. demonstrated that women with MME have uterine dysperistalsis during the late follicular phase which may compromise rapid sperm transport [30], although co-existent adenomyosis may be of greater importance in this context [31].

Various abnormalities of ovarian follicular development have been demonstrated in women with MME which compromise ovulation and impair the oocyte fertilizing potential [32]. These include a prolonged follicular phase [33], a reduced follicular growth rate [34], an impaired LH surge, and altered patterns of oestrogen and progesterone secretion [32]. The luteinized unruptured follicle (LUF) may be a cause of infertility in patients with MME [35]. Experimental animal models have demonstrated an alteration in the oocyte cytoskeleton induced by the peritoneal fluid of infertile women with endometriosis [36], with the findings of Gianaroli et al.’s prospective study supporting a higher risk of aneuploid gametes formation in women with endometriosis-associated infertility [37].

Women with endometriosis may also have reduced endometrial receptivity, as evidenced by the reduced expression of various endometrial receptivity markers [38]: Reduced endometrial expression of the avb integrin (a molecule important for embryo attachment at the time of implantation) has been demonstrated in women with endometriosis [39], as well as reduced levels of biomarkers of implantation, namely glycodelin A (GdA), osteopontin (OPN), lysophosphatidic acid receptor 3 (LPA3), and HOXA10 [40]. Furthermore, the presence of IgG and IgM anti-endometrial antibodies might be partially responsible for implantation failure in women with endometriosis [41,42]. Altered gene expression profiling leading to progesterone resistance in the eutopic endometrium of women with endometriosis has also been demonstrated, resulting in an unopposed oestrogen state, which is likely not favourable for implantation [43,44]. Qiao’s retrospective study found that 24% of infertile women with MME had chronic endometritis, which was linked with a lower cumulative pregnancy rate and live birth rate [45]. However, a review of oocyte donation studies concluded that women who receive oocytes from donors with endometriosis have lower implantation and pregnancy rates irrespective of the status of the recipient, suggesting that reduced fertility may be the result of impaired oocyte quality, rather than defective implantation [46].

### 2.2. Laparoscopic Surgery to Enhance Fecundity in Patients with Minimal/Mild Endometriosis

The logic behind surgically managing MME in order to enhance fecundity would be to minimize the potentially deleterious effect of peritoneal endometriosis implants on oocyte quality and/or implantation, by removing or destroying them. Indeed, Monsanto et al. demonstrated that the surgical removal of endometriosis lesions reduces local and systemic inflammation caused by the disease [47], which, as clarified earlier in the text, likely plays a central role in MME-associated infertility. Should adhesions co-exist with MME, the surgeon should aim to perform adhesiolysis at the same time, with a view to restoring normal pelvic anatomy, as adhesions may limit fallopian tube and/or ovarian mobility and also negatively impact fertility. The laparoscopic route (traditional laparoscopy or robot-assisted laparoscopy) should be preferred to laparotomy as it offers improved lesion visualization together with the well known benefits of minimal access surgery, namely, better recovery, less pain and improved cosmesis [48].

Regarding the role of surgery in the treatment of endometriosis-associated infertility, the most recent guideline by the European Society of Human Reproduction and Embryology (ESHRE) makes a weak recommendation that operative laparoscopy can be considered for rASRM stage I/II endometriosis as it improves the rate of natural, ongoing pregnancy, although the Guideline Development Group (GDG) acknowledges the lack of data on live birth rates, as well as the lack of a comparison with medically assisted reproduction (MAR) outcomes [22]. This recommendation is largely based on the findings of a Cochrane review by Bafort et al. [21], which included moderate-quality data from 3 randomised controlled trials (RCTs) and identified laparoscopic surgery to increase the rates of viable intrauterine pregnancy confirmed by ultrasound, compared to diagnostic laparoscopy only (OR 1.89; 95%CI 1.25 to 2.86): The first of those 3 RCTs was published as a conference abstract [49], including 41 infertile patients with stage I/II endometriosis, of which 20 underwent the resection or ablation of visible endometriosis lesions (group I), while the remaining 21 underwent diagnostic laparoscopy only (group II). Post-operative follow-up lasted up to 18 months after surgery or up to 20 weeks of gestation. Twenty-eight per cent of patients conceived in group I versus 23.8% in group II, with no reported cases of pregnancy loss in either group. Superovulation with 3 cycles of intrauterine insemination (IUI) was performed in women who failed to conceive after laparoscopy, resulting in an additional 5 pregnancies in group I and 4 in group II. The second study was a multi-centre, Canadian RCT (ENDOCAN) of 172 infertile patients who underwent ablation or excision of MME, and 169 infertile patients with MME who underwent diagnostic laparoscopy only [20]. The study analysed pregnancies that occurred up to 36 weeks post-operatively and proceeded to 20 weeks of gestation. The authors observed a near doubling of the cumulative pregnancy rates in the operative laparoscopy group (30.7% versus 17.7% in the diagnostic laparoscopy group, *p* = 0.006); however, the rates in both groups were low which suggests that other factors may have contributed as well. It should also be noted that a similar cumulative probability of pregnancy may be observed after a single in vitro fertilisation (IVF) attempt. Monthly fecundity rates per 100 person-months were 4.7% for the operative laparoscopy group and 2.4% in the diagnostic group. The absolute increase in the 36-week probability of a pregnancy carried beyond 20 weeks that was attributable to surgery was 13%, with no differences between the ablation and excision approach. The third RCT of the aforementioned Cochrane review [21], included 76 infertile patients with MME, half of whom were randomly allocated to undergo operative laparoscopy and half diagnostic laparoscopy only [50]: No significant difference was observed in the post-operative, natural conception rate between the 2 groups (only cases of spontaneous conception were included, as patients who received medical treatment to conceive after surgery were excluded from the study), during the 9-month follow-up (a 24% pregnancy rate in the operative laparoscopy group, compared with 18% in the diagnostic laparoscopy group, *p* = 0.49).

Similarly, an Italian multi-centre RCT (GISE study) failed to identify a benefit of laparoscopic ablation or resection of MME in terms of fecundity in infertile patients, compared to diagnostic laparoscopy only [51]: Both pregnancy rates (24% in the resection/ablation and 29% in the no-treatment group) and birth rates (20% in the resection/ablation group and 22% in the no-treatment group) during 1 year of follow-up were not significantly different between the 2 groups, regardless of whether post-operative medical therapy was used or not.

An important consideration is that of the number needed to treat (NNT), referring to the actual number of laparoscopies for MME needed to be performed, in order to achieve an additional pregnancy. Vercellini et al. identified NNT in this situation to be at least 12 [52]. However, if we take into account the fact that MME is not easily diagnosed pre-operatively and that around 30% of women with unexplained infertility may actually have MME [53], the NNT may increase to 40 [54]. On the other hand, the reported success rate of IVF is in the region of 25%, which corresponds to a NNT of around 4 [55]. Based on those figures, the Endometriosis Treatment Italian Club (ETIC) formulated a strong recommendation for clinicians not to perform surgery with the aim of diagnosing and treating superficial endometriosis in otherwise asymptomatic (without pelvic pain) infertile patients [56]. On the contrary, the co-existence of infertility with pelvic pain justifies performing operative laparoscopy with a view to enhancing fertility as well as alleviating symptoms [57].

Kalaitzopoulos et al. reviewed and compared 6 national and 2 international guidelines on endometriosis [58], and observed that the College National des Gynecologues et Obstetriciens Francais (CNGOF) [59], the National Institute for Health and Care (NICE) [60], the World Endometriosis Society (WES) [61], the National German Guideline (S2k) [62], American College of Obstetricians and Gynaecologists (ACOG) [63], and American Society for Reproductive Medicine (ASRM) [54], agree that, for patients with suspected MME and infertility, surgical management should be considered.

A recent network meta-analysis identified that pregnancy rates significantly increased following operative laparoscopy for endometriosis compared with placebo (odds ratio (OR) 1.63; 95%CI 1.13 to 2.35) [64]. However, the authors did not differentiate the outcomes between different endometriosis stages, and data on live birth rates were limited. Jin’s meta-analysis of 4 trials on MME identified laparoscopic surgery to increase live birth rates (relative risk (RR) 1.52, 95% confidence interval (CI) 1.26–1.84, *p* < 0.01) and pregnancy rates (RR of 1.44, 95% CI 1.24–1.68, *p* < 0.01) [65]. Comparable post-operative pregnancy rates, between 76% and 86% across all 4 stages of endometriosis, were reported in a recent retrospective study utilising a combined hystero-laparoscopy approach with CO_2_ laser, with the authors concluding that the stage of the disease does not impact the post-operative fertility outcome [66].

Regarding the optimal surgical technique (ablation versus excision of MME lesions) in order to enhance fertility, there is no robust evidence to suggest superiority of one approach over the other. However, radical excision of all affected peritoneum with a sufficient safety margin is an attractive approach for the adequately skilled surgeon, as it may be linked with post-operative pregnancy rates in excess of 60% in infertile patients, according to one retrospective study [67]. As with surgery for any stage/phenotype of endometriosis, post-operative recurrence remains a major concern, particularly in patients seeking pregnancy after surgery, for whom hormonal contraception is not a suitable option. The actual risk of recurrence for MME may be harder to accurately estimate compared to a more advanced disease, owing to the well known limitations of non-invasive diagnosis of MME; however, this has been estimated to be as high as 21.5% (2 years after surgery) and 40–50% (5 years after surgery) [68].

Another point to examine is the role of medical therapy before the surgical management of MME. The use of hormones pre-operatively to suppress inflammation secondary to endometriosis makes sense; however, there is the theoretical risk of making some MME lesions invisible at the time of laparoscopy, thereby resulting in incomplete treatment [69]. Furthermore, its clinical benefit has not been demonstrated [70]. A Cochrane review concluded that the use of ovulation suppression agents does not confer any benefit in infertile women with endometriosis wishing to conceive [71], and, understandably, the latest ESHRE guidance does not recommend their use for this purpose [22].

As regards the post-operative administration of GnRH agonists in women with MME, Söritsa et al. did not identify any beneficial effect on spontaneous or IVF pregnancy rates [72]. Decleer et al. focused on IVF pregnancies and did not identify any benefit of adding a 3-month down-regulation with GnRH agonist prior to the conventional long IVF protocol in MME patients who had been managed laparoscopically with a laser. On the contrary, patients who received laser laparoscopy for MME followed by conventional long IVF protocol (controls) required lower doses of FSH and a shorter duration of stimulation with no difference in the number of metaphase II (MII) oocytes or pregnancy rate [73]. Kaponis’s RCT found that pre-treatment with a GnRH agonist in patients with laparoscopically managed MME (bipolar cautery) planned to undergo IVF improves the fertilisation rate but not the clinical pregnancy rate, whilst reducing concentration of cytokines in the follicular fluid [74].

An RCT examined the role of post-operative down-regulation with GnRH agonist before COS/IUI in patients who had been surgically managed for MME: Pregnancy rates and live birth rates did not differ between those who received GnRH agonist and those who did not [75]. However, a more recent retrospective study identified that adding a GnRH agonist post-operatively, prior to COS/IUI, led to a higher clinical pregnancy rate (15.29% vs. 11.82% in controls, *p* = 0.035) and a, non-statistically significant, higher live birth rate (12.94% vs. 10%, *p* = 0.311) [76].

Regarding the use of post-operative COS in patients with MME, Boujenah et al.‘s retrospective study observed an improvement in pregnancy rates by using recombinant or urinary gonadotrophins, whereas the addition of IUI did not confer additional benefit [77]. In comparing letrozole versus clomiphene as COS agents in women who underwent IUI within 6 to 12 months after surgery for MME, Abu Hashim’s RCT identified comparable cumulative pregnancy rates [78]. In Alborzi’s retrospective study, post-operative pregnancy rates did not differ significantly between patients who underwent laparoscopic management followed by letrozole for 2 months (pregnancy rate of 23.4%), those who received triptorelin for 2 months after surgery (pregnancy rate of 27.5%) and those who did not receive either following surgery (pregnancy rate of 28.1%) [79]. According to ESHRE guidance, the use of COS/IUI within 6 months after surgical management of stage I/II endometriosis may be considered [22].

### 2.3. Laparoscopy versus Other Modalities in Patients with Minimal/Mild Endometriosis

Although operative laparoscopy for MME does not carry the operative risks of more advanced disease, surgical and/or anaesthetic risks may discourage certain patients from this intervention. Expectant management may seem an attractive alternative as 50% of patients with MME will eventually conceive spontaneously without intervention [80]. Available studies report contradictory results on the probability of achieving a pregnancy amongst those with unexplained infertility and those with stage I/II endometriosis. Berube et al. identified spontaneous pregnancy rates in cases of stage I/II endometriosis to be comparable with those of unexplained infertility [81]. However, Akande et al. observed that patients with MME have a lower probability of pregnancy over 3 years, compared to those with unexplained infertility and a normal pelvis (36% vs. 55%, respectively) [19]. It should be remembered that an overlap between cases classified as unexplained infertility and MME is likely to exist in the literature, as a recent systematic review found 44% of women with unexplained infertility to have endometriosis, of which 74% was SUP [82].

Another reasonable alternative may be controlled ovarian stimulation (COS) with in-utero insemination (IUI). Indeed, the most recent ESHRE guideline makes a weak recommendation that, in infertile women with rASRM stage I/II endometriosis, IUI with COS, instead of expectant management or IUI alone, may be performed as it increases pregnancy rates [22]. Tummon’s RCT demonstrated that COS with FSH and IUI achieved significantly higher live birth rates, compared to the expectant management of infertile women with MME (11% versus 2%) [83]. Indeed, the addition of COS with clomiphene citrate or gonadotrophins to IUI was demonstrated to improve outcomes in patients with endometriosis compared to IUI alone in a large multi-centre cohort study. However, the authors did not differentiate between stages of endometriosis [84]. Patients with early-stage endometriosis appear to have lower clinical pregnancy rates following COS/IUI compared to those suffering from unexplained infertility, according to certain studies [85,86,87]. However, in Isaksson’s study, the difference was not significant (27.7% pregnancy rate in unexplained infertility versus 18.4% in MME) [88]. Furthermore, following the surgical treatment of stage I/II endometriosis, the pregnancy rate per therapy cycle and cumulative live birth rate are comparable to patients with unexplained infertility [89], indicating a detrimental effect of endometriosis per se on the fertility outcome.

Another interesting point would be to compare the outcomes of operative laparoscopy versus medical management. In Milingos’s prospective study [90], the cumulative probability of pregnancy rates did not differ significantly between infertile patients who underwent ablation/resection of MME (group 1) and those with diagnostic laparoscopy followed by 6 months of GnRH agonist (group 2) (*p* = 0.19); however, the rates were significantly higher in both groups compared to diagnostic laparoscopy only (group 3). The post-treatment cumulative intrauterine pregnancy rates during the 24-month follow-up period were 36.7% (group 1), 30.5% (group 2) and 20.9% (group 3).

### 2.4. The Impact of Minimal/Mild Endometriosis on IVF Outcomes

In Harb et al.’s meta-analysis of the impact of endometriosis on in vitro fertilization (IVF) outcomes [91], data from 7 studies that reported the fertilization rate as an outcome for stage I/II endometriosis identified that stage I/II endometriosis was associated with a 7% reduction in the fertilization rate, but that there was no difference in implantation, clinical pregnancy, and live birth rate compared to controls. Despite a 21% reduction in pregnancy rates in patients with severe endometriosis, no significant difference in live birth rate was observed between patients with MME and those with severe disease. The negative impact of MME on the fertilization rate had been demonstrated by an earlier meta-analysis which found that fertilization rates were lower for MME compared to severe endometriosis or tubal factor infertility [92]. A more recent meta-analysis confirmed that MME specifically impairs fertilization (OR 0.77, CI 0.63–0.93) and earlier implantation processes (OR 0.76, CI 0.62–0.93), whereas more severe endometriosis impacts negatively on all reproduction stages [93]. In Barbosa et al.’s meta-analysis, patients with ΜΜΕ had similar clinical pregnancy and live birth rates to patients with other causes of infertility, as well as to patients suffering from stage III/IV endometriosis. In particular, the clinical pregnancy rate was 38% in stage I/II, and 34.2% in stage III/IV (RR: 0.90, 95%CI: 0.82–1.0), while the live birth rate was 28.2% in stage I/II, and 26.5% in stage III/IV (RR: 0.94, 95%CI: 0.80–1.11) [94]. Rossi’s meta-analysis found that patients with stage I/II endometriosis undergoing IVF had a similar clinical pregnancy rate to controls [95]. Regarding live birth rates, a meta-analysis concluded that patients with stage I/II endometriosis had comparable live birth rates following IVF compared to patients without endometriosis [96]. In particular, the live birth rates in eight studies (4157 patients) had OR 0.96, 95% CI 0.82–1.12; the clinical pregnancy rate in 15 studies (9692 patients) had OR 0.84, 95% CI 0.69–1.03; and the mean number of oocytes retrieved per cycle in 11 studies was (mean difference –0.58, 95% CI: 21.16 to 0.01). The exception was for patients with moderate/severe disease who had 30% lower live birth and 40% lower clinical pregnancy rates. Excluding a single retrospective study that compared the surgical ablation of MME to diagnostic laparoscopy only prior to IVF [97], Hamdan et al. identified that, in the subgroup of patients with stage I/II endometriosis, the live birth rate, the clinical pregnancy rate and the mean number of oocytes retrieved per IVF cycle did not differ between those who had stage I/II surgically managed prior to IVF and those where surgical treatment status was not specified [96]. A large retrospective-cohort study observed that infertile women with a diagnosis of isolated endometriosis had similar or higher live birth rates compared with those with unexplained infertility (RR = 1.04), tubal factor (RR = 1.04) or all other diagnoses (RR = 1.1). However, in patients where endometriosis co-exists with other alterations in the genital tract, both implantation rates and live birth rates were lower compared with unexplained infertility, tubal factor, and all other diagnostic groups [98]. Regarding the role of endometrial receptivity in endometriosis-associated infertility, clinical and live birth rates in frozen embryo transfer (FET) cycles of euploid embryos did not differ between patients with infertility and controls without suspected endometrial factors [99,100]. However, Bishop et al. did not differentiate between stages of endometriosis [99].

The latest ESHRE guidance does not recommend prolonged GnRH agonist or combined contraceptive/progestogen use before planned IVF, as evidence that this approach increases live birth rates is lacking [22].

### 2.5. The Role of Laparoscopic Management of Minimal/Mild Endometriosis Prior to ART

As regards the role of surgery in enhancing ART outcomes in women with MME, the most recent ESHRE guideline does not recommend that clinicians routinely perform surgery prior to ART in order to enhance live birth rates [22]. This is, in the main, on the basis of a lack of high-quality evidence on the potential benefits of surgery and is reflected in the CNGOF guideline [59], as well as in published expert commentary [101]. Indeed, the only original study to compare the surgical ablation of MME prior to ART with diagnostic laparoscopy followed by ART, was a retrospective study by Opoien et al. which linked the former approach with improved reproductive outcomes [97]. In particular, in this large retrospective cohort study of 661 infertile women with MME and more than 1600 IVF cycles in total, those who underwent ablation of laparoscopic lesions prior to the first IVF cycle (*n* = 399) experienced a significantly improved implantation rate (30.9% versus 23.9%, *p* = 0.02), pregnancy rate (40.1% versus 29.4%, *p* = 0.004), live-birth rate per ovum retrieval (27.7% versus 20.6%, *p* = 0.04), and a shorter time to 1st pregnancy, compared to those who underwent diagnostic laparoscopy only (*n* = 262). It is worth noting that the same group of authors, in a subsequent retrospective study, identified that IVF and ICSI success rates were similar between patients with various endometriosis stages (excluding ovarian endometriomas) and those with tubal factor infertility [102]. A very recent meta-analysis on this topic identified that surgery for endometriosis before IVF does not impact the ongoing pregnancy rate {1.28 [0.66, 2.49]; I^2^ = 60%; *n* = 3} or early pregnancy loss rate {0.88 [0.62, 1.25]; I^2^ = 0%; *n* = 7} per cycle; however, after excluding studies with a high risk of bias, the live birth rate was lower for those who underwent surgery pre-IVF than for those who did not [103].

### 2.6. The Future Role of Non-Invasive Diagnosis in the Management of Minimal/Mild Endometriosis-Related Infertility

The development of a reliable, widely available and, ideally, inexpensive test for the non-invasive diagnosis of MME has long been desired in order to reduce the need for diagnostic laparoscopies which, as invasive procedures, are related to morbidity, or even mortality [104]. A Cochrane review identified no reliable biomarkers in the blood for clinical use in the diagnosis of endometriosis [105], with similar conclusions drawn for urinary biomarkers [106]. The latest ESHRE guidance does not recommend that clinicians measure biomarkers in blood, urine, menstrual fluid or endometrial tissue in order to diagnose endometriosis [22]. Micro-RNAs (miRNAs) have attracted considerable attention, and a recent prospective study using next-generation sequencing and artificial intelligence identified a salivary miRNA signature consisting of 89 miRNAs, specific to SUP, with high diagnostic accuracy [107]. Further solid, scientific evidence is eagerly awaited before routinely recommending this test.

Given the reported high prevalence of MME in patients with unexplained infertility [53,82], and the reported positive impact of the laparoscopic management of MME on fecundity and live birth rates [20,21,22], it is the authors’ view that the development of a reliable, non-invasive diagnostic tool for MME is likely to revolutionise the management of couples currently classified as having unexplained infertility, as it will allow for the laparoscopic management of those patients diagnosed with MME, potentially reducing the need for the use of medically-assisted reproduction. Consequently, an anticipated rise in the number of operative laparoscopies for MME should not, in the authors’ view, be regarded with scepticism but, rather, as an approach to maximize the patient’s reproductive potential.

## 3. Conclusions

Laparoscopic management of MME enhances fecundity and increases the chances of spontaneous conception in appropriately selected cases. This intervention allows for the concurrent lysis of potentially co-existent adhesions that may also have a negative impact on fertility. However, based on the currently available evidence, routine laparoscopy should not be performed in asymptomatic, infertile women with the sole aim of diagnosing and managing potentially present MME. Similarly, laparoscopic management of MME cannot be routinely recommended prior to ART in order to solely enhance fertility outcomes, as strong evidence of a beneficial effect is lacking and is based on a single retrospective study. Limited available evidence suggests that COS/IUI might improve pregnancy rates following the laparoscopic management of MME. Surgical and anaesthetic risks should be taken into consideration and discussed in depth with the patient when surgery is contemplated.

Providing solid evidence on the impact of MME (and its surgical management) on fertility outcomes remains a challenging task due to factors pertaining to the heterogeneity of data; the co-existence of different phenotypes of endometriosis; authors using different staging systems; not differentiating between disease stages; not reporting outcomes separately for each stage; and different surgical approaches (ablation versus excision), as well as the presumably high number of missed diagnoses of MME owing to the lack of reliable, non-invasive diagnostic tools. In a disease as complex and heterogeneous as endometriosis, every clinical decision should be individualised, taking into account factors such as the patient’s preference, age, the co-existence of pain symptoms, ovarian reserve and past surgical history, as well as the healthcare costs of alternatives to surgery. As with any decision that pertains to endometriosis surgery, if surgery is to be performed, it should be carried out by appropriately trained clinicians with the aim of fully eradicating/destroying the disease whilst, at the same time, minimising surgical trauma that may lead to further adhesion formation. Future studies should focus on reporting the effects of certain interventions for MME on fertility and obstetric outcomes, clearly differentiating between disease stages. Given the relatively high incidence of MME in patients classified as having unexplained infertility, the much anticipated development of a reliable, non-invasive diagnostic tool is likely to significantly impact the decision-making process of performing laparoscopic surgery for these patients.

## Data Availability

No new data were created or analysed in this study. Data sharing is not applicable to this article.

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
