# Peer review of "The Impact of Laparoscopic Surgery on Fertility Outcomes in Patients with Minimal/Mild Endometriosis"

_jcm, 2024, doi:10.3390/jcm13164817_

Round 1

Reviewer 1 Report

Comments and Suggestions for Authors

Comments

1.       LINE 50 Endometriosis may, also, present as pelvic peritoneal defects (Allen-Masters syndrome) [13], while histological evidence of endometriosis (occult endometriosis) may be found in biopsies of clinically negative peritoneum from patients with pelvic pain [14)- I THINK IT IS IMPORTANT TO MENTION HERE THAT THE CLINICAL SIGNIFICANCE OF THESE CLINICALLY NEGATIVE PERITONEAL BIOPSIES IN RESOLUTION OF PAIN OR INFERTILITY IS UNKNOWN.

2.      Line 103 ‘. Qiao’s retrospective study found that 24% of infertile women with MME had chronic endometritis, which was linked with lower cumulative pregnancy rate and livebirth rate [45].’ THIS REQUIRES A QUALIFIER AS 24 % PREVALENCE HAS NOT BEEN CONFIRMED

3.     Line 198 ‘Regarding the optimal surgical technique (ablation versus excision of MME lesions) in order to enhance fertility, there is no robust evidence to suggest superiority of one approach over the other’. THE AUTHORS NEED TO MAKE IT CLEARER THAT FOR EARLY STAGE DISEASE THERE IS NO EVIDENCE THAT EXCISION IS BETTER- REFERENCES ARE PLENTY ( EX. Excision versus Ablation for Management of Minimal to Mild Endometriosis: A Systematic Review and Meta-analysis. Burks C, Lee M, DeSarno M, Findley J, Flyckt R.J Minim Invasive Gynecol. 2021 Mar;28(3):587-597)

4.     I SUGGEST THIS ARTICLE ( NOT MINE) WHERE THEY REVIEW THAT ALL PREGNANCY PARAMETERS ARE THE SAME IN FET BWHETHER YOU TREAT OR DON’T TREAT ENDOMETRIOSIS -
Effects of endometriosis on assisted reproductive technology: gone with the wind.
Pirtea P, de Ziegler D, Ayoubi JM.Fertil Steril. 2021 Feb;115(2):321-322

Author Response

Comment 1:  LINE 50 Endometriosis may, also, present as pelvic peritoneal defects (Allen-Masters syndrome) [13], while histological evidence of endometriosis (occult endometriosis) may be found in biopsies of clinically negative peritoneum from patients with pelvic pain [14)- I THINK IT IS IMPORTANT TO MENTION HERE THAT THE CLINICAL SIGNIFICANCE OF THESE CLINICALLY NEGATIVE PERITONEAL BIOPSIES IN RESOLUTION OF PAIN OR INFERTILITY IS UNKNOWN.

Response 1:Thank you kindly for your pertinent comment. We have made the relevant change in the updated manuscript version to reflect this. We added: ‘’, although the clinical significance of occult endometriosis in the context of pain or infertility remains unclear.’’, (line 58 and 59).

Comment 2:  Line 103 ‘. Qiao’s retrospective study found that 24% of infertile women with MME had chronic endometritis, which was linked with lower cumulative pregnancy rate and livebirth rate [45].’ THIS REQUIRES A QUALIFIER AS 24 % PREVALENCE HAS NOT BEEN CONFIRMED.

Response 2: Thank you but we are not entirely certain what you mean here. We have rechecked the paper and confirmed that the authors report the prevalence to be 24%.

Comment 3: Line 198 ‘Regarding the optimal surgical technique (ablation versus excision of MME lesions) in order to enhance fertility, there is no robust evidence to suggest superiority of one approach over the other’. THE AUTHORS NEED TO MAKE IT CLEARER THAT FOR EARLY STAGE DISEASE THERE IS NO EVIDENCE THAT EXCISION IS BETTER- REFERENCES ARE PLENTY ( EX. Excision versus Ablation for Management of Minimal to Mild Endometriosis: A Systematic Review and Meta-analysis. Burks C, Lee M, DeSarno M, Findley J, Flyckt R.J Minim Invasive Gynecol. 2021 Mar;28(3):587-597). 

Response 3: Thank you but we politely disagree with your comment. We feel that our phrasing reflects the fact that no technique is superior to the other in terms of fertility outcomes. We have read your suggested reference and this systematic review and meta-analysis examines the outcomes in terms of pain and not fertility. Even when this is considered, the authors of this systematic review and meta-analysis conclude by stating: ‘’ Although the data from our systematic review and pooled meta-analysis demonstrate no significant difference between laparoscopic excision and ablation in regard to improving postoperative pain from minimal to mild endometriosis, we believe that strong and definitive conclusions cannot be made at this time on the basis of the literature that is currently available’’.

Comment 4: I SUGGEST THIS ARTICLE ( NOT MINE) WHERE THEY REVIEW THAT ALL PREGNANCY PARAMETERS ARE THE SAME IN FET BWHETHER YOU TREAT OR DON’T TREAT ENDOMETRIOSIS -
Effects of endometriosis on assisted reproductive technology: gone with the wind. Pirtea P, de Ziegler D, Ayoubi JM.Fertil Steril. 2021 Feb;115(2):321-322.

Response 4: Thank you, we have indeed included this reference as well as the original study by Bishop et al. in our updated manuscript. We write: ‘’Regarding the role of endometrial receptivity in endometriosis-associated infertility, clinical and livebirth rates in frozen embryo transfer (FET) cycles of euploid embryos did not differ between patients with infertility and controls without suspected endometrial factors [100,101], however, Bishop et al. did not differentiate between stages of endometriosis [100].’’, line 334-338.

Reviewer 2 Report

Comments and Suggestions for Authors

A very nice review. Detailed and informative, so it could be suitable for a book chapter, not just a journal article. The organization of the work is excellent, from the introduction about endometriosis, its general prevalence and impact on infertility, classification, clinical picture, and pathogenesis of the disease. The section on laparoscopy as a method of treating endometriosis is also good, supported by reference studies. My only criticism is of section 2.5, where the authors discuss the significance of laparoscopy in treating endometriosis before the IVF procedure; this part of the work is short. I would ask the authors to supplement it with more relevant studies in this field. The conclusion is good and concise.

Author Response

Comment 1: A very nice review. Detailed and informative, so it could be suitable for a book chapter, not just a journal article. The organization of the work is excellent, from the introduction about endometriosis, its general prevalence and impact on infertility, classification, clinical picture, and pathogenesis of the disease. The section on laparoscopy as a method of treating endometriosis is also good, supported by reference studies. My only criticism is of section 2.5, where the authors discuss the significance of laparoscopy in treating endometriosis before the IVF procedure; this part of the work is short. I would ask the authors to supplement it with more relevant studies in this field. The conclusion is good and concise.

Response 1: Thank you kindly for your comments. The reason as to why this section is short is because of the limited evidence published. We have been unable to identify any other studies pertinent to this topic, therefore, we respectfully do not feel that there is a need to make this section longer. However, following your comment, we have identified an error in this section of the original text and we have corrected it: in line 363, the word ‘’higher’’ has been replaced by ‘’lower’’.

Reviewer 3 Report

Comments and Suggestions for Authors

The manuscript appears clear and relevant to the field of reproductive medicine and surgery, particularly in relation to the treatment of endometriosis-related infertility. It provides detailed insights into the benefits and limitations of laparoscopic treatment of minimal to mild endometriosis (MME) to improve fertility. Discussing the heterogeneity of data, the complexity of the condition and the need for individualized clinical decisions based on various patient-specific factors, it presents a well-structured and comprehensive account. In addition, the importance of appropriately trained clinicians to perform surgery is emphasized and the need for future research and development of non-invasive diagnostic tools is highlighted.

The manuscript discusses the impact of laparoscopic treatment of MME on fertility outcomes based on currently available data. It does not present new experimental data or a specific experimental design, but rather synthesizes and evaluates existing research.

The manuscript appears scientifically sound as it critically analyzes the current evidence on the efficacy of laparoscopic treatment of MME to increase fertility.

- It acknowledges the limitations of existing data, such as heterogeneity, different endometriosis phenotypes and different surgical approaches.

- It also emphasizes the need for more rigorous studies to provide solid evidence of the effects of MME treatment on fertility.

There are no figures or schematics to facilitate acceptance of the information provided with this review, and I strongly recommend that the authors provide them to highlight the most important aspects of the review.

Overall, the conclusions follow logically from the evidence and arguments discussed, highlighting both the potential benefits and limitations of laparoscopic treatment of MME to improve fertility.

Author Response

Comment 1: 

The manuscript appears clear and relevant to the field of reproductive medicine and surgery, particularly in relation to the treatment of endometriosis-related infertility. It provides detailed insights into the benefits and limitations of laparoscopic treatment of minimal to mild endometriosis (MME) to improve fertility. Discussing the heterogeneity of data, the complexity of the condition and the need for individualized clinical decisions based on various patient-specific factors, it presents a well-structured and comprehensive account. In addition, the importance of appropriately trained clinicians to perform surgery is emphasized and the need for future research and development of non-invasive diagnostic tools is highlighted.

The manuscript discusses the impact of laparoscopic treatment of MME on fertility outcomes based on currently available data. It does not present new experimental data or a specific experimental design, but rather synthesizes and evaluates existing research.

The manuscript appears scientifically sound as it critically analyzes the current evidence on the efficacy of laparoscopic treatment of MME to increase fertility.

- It acknowledges the limitations of existing data, such as heterogeneity, different endometriosis phenotypes and different surgical approaches.

- It also emphasizes the need for more rigorous studies to provide solid evidence of the effects of MME treatment on fertility.

There are no figures or schematics to facilitate acceptance of the information provided with this review, and I strongly recommend that the authors provide them to highlight the most important aspects of the review.

Overall, the conclusions follow logically from the evidence and arguments discussed, highlighting both the potential benefits and limitations of laparoscopic treatment of MME to improve fertility.

Response 1: Thank you kindly for your comments. Following your suggestion, and in line with the journal’s policy, we have created a graphical abstract that aims to summarize the key points of our review. This has been submitted, together with the updated manuscript.